# Serum Fatty Acids and Inflammatory Patterns in Severe Obesity: A Preliminary Investigation in Women

**DOI:** 10.3390/biomedicines12102248

**Published:** 2024-10-03

**Authors:** Gislene B. Lima, Nayra Figueiredo, Fabiana M. Kattah, Emilly S. Oliveira, Maria A. Horst, Ana R. Dâmaso, Lila M. Oyama, Renata G. M. Whitton, Gabriel I. M. H. de Souza, Glaucia C. Lima, João F. Mota, Raquel M. S. Campos, Flávia C. Corgosinho

**Affiliations:** 1Postgraduate Program in Nutrition and Health, Faculty of Nutrition, Federal University of Goiás (UFG), Goiânia 74690-900, Brazil; fabianakattah@discente.ufg.br (F.M.K.); emillysantos@discente.ufg.br (E.S.O.); aderuza@ufg.br (M.A.H.); glauciacarielo@ufg.br (G.C.L.); jfemota@gmail.com (J.F.M.); flaviacorgosinho@ufg.br (F.C.C.); 2Postgraduate Program in Health Sciences, Faculty of Medicine, Federal University of Goiás (UFG), Goiânia 74690-900, Brazil; nayra_figueiredo@discente.ufg.br; 3Interdisciplinary Postgraduate Program in Health Sciences, Federal University of São Paulo (UNIFESP), São Paulo 04039-032, Brazil; ana.damaso@unifesp.br (A.R.D.); lmoyama@unifesp.br (L.M.O.); gabrinacio@gmail.com (G.I.M.H.d.S.); raquelmunhoz@hotmail.com (R.M.S.C.); 4Institute of Biosciences, Federal University of São Paulo (USP), São Paulo 05508-900, Brazil; renata.fish@gmail.com

**Keywords:** leptin, adiponectin, omega-3 fatty acid, omega-6 fatty acid, inflammation

## Abstract

**Background:** Inflammation plays a central role in many chronic diseases that characterize modern society. Leptin/adiponectin and adiponectin/leptin ratios have been recognized as notable markers of dysfunctional adipose tissue and, consequently, an inflammatory state. **Methods:** Blood samples were collected from 41 adult volunteers (40.2 ± 8.3 years) diagnosed with severe obesity (BMI 46.99; 42.98–51.91 kg/m^2^). The adipokines were quantified using an enzyme-linked immunosorbent assay, while the serum fatty acid analysis was conducted using chromatography. **Results:** The results unveiled a positive correlation between the leptin/adiponectin ratio and the 20:3n6 fatty acid (r = 0.52, *p* = 0.001), as well as a similar positive correlation between the adiponectin/leptin ratio and the 22:6n3 fatty acid (r = 0.74, *p* = 0.001). In the regression analysis, the 22:6n3 fatty acid predicted the adiponectin/leptin ratio (β = 0.76, *p* < 0.001), whereas C20:3 n-6 was a predictor for inflammatory markers (β = 4.84, *p* < 0.001). **Conclusions:** In conclusion, the 22:6n3 fatty acid was demonstrated to be a predictive factor for the adiponectin/leptin ratio and C20:3 n-6 was a predictor for inflammatory markers. This discovery, novel within this population, can help develop new intervention strategies aimed at controlling the inflammatory status in individuals classified as having severe obesity.

## 1. Introduction

Obesity is a growing global public health concern, characterized by the excessive expansion of adipose tissue, which contributes to a pro-inflammatory state [1,2]. Individuals with obesity often exhibit high leptin concentrations and adipokines with pro-inflammatory properties. In fact, studies have demonstrated that high leptin levels are related to cardiometabolic dysfunction [3,4] in both the adult and pediatric populations [5,6,7], highlighting the importance of this marker in obesity and its related diseases. Conversely, adiponectin is reduced in individuals with obesity, playing an anti-inflammatory, anti-diabetic, and cardioprotective role, underscoring the clinical relevance of hypoadiponectinemia in this population [2,8,9,10,11]. Considering that leptin and adiponectin interact with each other in the modulation of cardiometabolic risk, recent studies have suggested that the leptin/adiponectin (lep/adipo) and adiponectin/leptin (adipo/lep) ratios may serve as more precise markers of pro- and anti-inflammatory states, respectively, compared to individual adipokines [12,13,14,15,16].

It has been demonstrated, in both human and experimental studies, that fatty acids might play a role in the pathogenesis of obesity, inflammation, and cardiometabolic disease. 

In fact, omega-3 fatty acids have been described as attenuating low-grade inflammation in the adipose tissue of obese individuals [2]. On the other hand, studies suggest that the excessive consumption of trans-saturated fatty acids and omega-6 fatty acids is associated with a pronounced inflammatory state [17,18]. Despite the recognized link between inflammation and fatty acids [19], no study has explored this correlation, particularly in the population with severe obesity, which poses considerably heightened risks.

Considering the increasing prevalence of obesity, the significance of inflammation in disease outcomes, and the potential role of fatty acids in the inflammatory process, there is a gap in the literature regarding this triad. Given the significantly increased risks of severe obesity for cardiometabolic health, especially in women, a better understanding of the relationship between inflammation markers and fatty acids may be crucial for effective obesity management. Therefore, this study aims to assess the association between the adipo/lep and lep/adipo ratios and serum fatty acids in women with severe obesity.

## 2. Materials and Methods

### 2.1. Design and Participants

This is a cross-sectional observational study approved by the Research Ethics Committee of the Federal University of Goiás and the Dr. Alberto Rassi State Hospital—HGG (protocol number 961/19). The sample consisted of women from the public health system awaiting bariatric surgery, who were recruited from July to December 2019 (the vast majority of patients undergoing bariatric surgery at HGG are women). All eligible volunteers provided informed consent and signed a form in duplicate before participating in the study. The inclusion criteria were women with severe obesity (BMI ≥ 40 kg/m^2^) aged between 18 and 59 years. The exclusion criteria included acute inflammatory diseases, infectious diseases, neoplastic diseases, or genetic syndromes; alcohol consumption (>30 g/day); or the use of illicit drugs.

### 2.2. Anthropometric Assessment

We conducted the measurements encompassing the height, body mass, and waist and hip circumferences. The body mass was determined using a Tanita-UM 080 digital scale with a maximum capacity of 150 kg. The height was measured using an inextensible metric, and the volunteers were asked to stand in an upright position. The waist circumference was measured at the midpoint between the last rib and the iliac crest, following a deep breath, with the volunteers standing. The hip circumference was measured at its maximum identified diameter, with the volunteers instructed to maintain their feet together during the assessment. All measurements were taken in duplicate by the same researcher staff using an inelastic tape. The BMI was calculated and classified according to the World Health Organization, dividing the body mass weight by the square of the height [20].

### 2.3. Analysis of Adipokines

The blood collection took place following a 12 h overnight fast. In triplicate, the serum samples were collected into vacuum tubes (Labor Import, Osasco, Brazil) after centrifugation (Eppendorf 5702R centrifuge, Hamburg, Germany) at 4000 RPM for 10 min at 10 °C and stored at −80 °C for subsequent analysis. ELISA kits (R&D Systems, Minneapolis, MN, USA) were employed for the quantification of leptin and adiponectin, following the manufacturer’s instructions.

### 2.4. Analysis of Free Fatty Acid Profile in Serum

The analysis of free fatty acid composition in serum was determined using gas chromatography with a Varian 3900 gas chromatograph (Walnut Creek, CA, USA) coupled with flame ionization detection (FID) and an automatic sampler CP-8410. The methylation of each fraction was performed using acetyl chloride (5% HCl and methanol), and the fatty acid composition was determined with methyl esters. Fatty acids were identified by comparing retention times using a known standard of fatty acid methyl ester (FAME). FAMEs were utilized on a capillary column (CP Wax 52 CB, Varian, Lake Forests, CA, USA) with a thickness of 0.25 µm, an inner diameter of 0.25 mm, and a length of 30 m. Hydrogen was used as the carrier gas at a linear velocity of 22 cm/s. The temperature was programmed to 170 °C for 1 min, followed by increments of 2.5 °C/min until 240 °C, with a final hold time of 5 min. The injector temperature was set at 250 °C, and the FID was set at 260 °C. FAMEs were identified by comparing the retention times of the samples with known standards (Supelco, 37 components; Sigma-Aldrich, St. Louis, MO, USA; Mixture, Me93, Larodan, and Qualmix, PUFA Fish M, Menhaden Oil, Larodan, Solna, Sweden). Percentages of total fatty acids were used to express the values of fatty acid composition.

### 2.5. Statistical Analysis

The statistical analysis was performed using the R Studio program, version 4.3.3. The normality of the variables was assessed using the Shapiro–Wilk test, and the data are presented as mean and standard deviation or median and quartile range, according to normality. To investigate the association between the leptin/adiponectin and adiponectin/leptin ratios and serum fatty acids, and also between C-reactive proteins (CRPs) and omega-3 and -6 fatty acids, Pearson or Spearman correlation analyses were performed based on data normality. A generalized linear regression model (GLM) was performed using the Stepwise strategy, with age, BMI, waist circumference, hip circumference, and neck circumference in the model. The covariates were chosen according to similar models found in the literature [21,22,23].

## 3. Results

Forty-nine patients were enrolled in the study; however, one person withdrew and seven were excluded due to a lack of data, such as the waist and hip circumferences (three volunteers) and adipokines measurements (four volunteers). Thus, the analyses were performed with 41 participants. The volunteers had a mean age of 40.2 ± 8.3 years and a BMI of 46.99 (42.97–51.90) kg/m^2^. A total of twenty-one plasma fatty acids, including four saturated (56.2% by area), five monounsaturated (17.2% by area), and twelve polyunsaturated fatty acids (26.6% by area), were identified. Regarding the adipokines, 60% of the sample presented hyperleptinemia and 41% presented hypoadiponectinemia. Data from the descriptive analysis of the sample are presented in Table 1.

The correlation analyses showed a positive correlation between the Lep/Adipo ratio and Di-homo-γ-linolenic acid (DGLA 20:3n6, r = 0.52, *p* = 0.00, Appendix A), as well as between the Adipo/Lep ratio and Docosahexaenoic Acid—DHA (22:6n3, r = 0.74, *p* = 0.00, Appendix A). Moreover, a positive correlation was found between CRP and 18:3n6 (*p* = 0.01), 20:4n3 (*p* = 0.00), and 18:3n-3 (*p* = 0.01) (Appendix A).

The multiple linear regression between C22:6n3 fatty acid and the Adipo/Lep ratio explains 63% of the variability of the ratio. And the regression between C20:3n6 and the Lep/Adipo ratio was adjusted for neck circumference and explains 30% of the variability of this ratio. The results showed C22:6n3 as a predictor of the Adipo/Lep ratio (*p* < 0.000) (Table 2). In addition, C20:3n6 is a predictor factor for the Lep/Adipo ratio (*p* < 0.000) (Table 3).

## 4. Discussion

The pro-inflammatory state in individuals with obesity is one of the main factors associated with the development of comorbidities and can be influenced by the circulation of fatty acids. It is already established in the literature that the analysis of the leptin/adiponectin and adiponectin/leptin ratios is a better indicator of inflammation than adipokines alone [12,14,15]. Considering the important role of inflammation in cardiovascular disorders, a more profound understanding of the relationship between these ratios and fatty acids can be of great value for the management of obesity and its comorbidities. For the first time, the present study demonstrated that the C22:6n3 fatty acid was a predictive factor for the increase in the Adipo/Lep ratio (β = 0.760, *p* < 0.000) in women with severe obesity. The regression model explains that 63% of the variability in the Adipo/Lep ratio is due to C22:6n3.

A previous study, comparing eutrophic individuals and individuals with obesity, identified DHA as a potential biomarker for chronic inflammation in obesity [24], corroborating our findings. Additionally, in a study with post-pubertal adolescents with obesity, it was observed that polyunsaturated fatty acids (PUFA) and, specifically, the n-3/n-6 ratio were positively correlated with adiponectin levels [19]. Finally, both experimental and clinical studies demonstrated a significant increase in adiponectin concentrations after n-3 fatty acid supplementation, reinforcing the positive influence of this type of fatty acid on the concentration of this adipokine [25].

The proposed mechanism for the influence of DHA on adiponectin concentration is related to its ability to act as a ligand for peroxisome proliferator-activated receptors (PPARs), which induces the expression of several genes involved in the metabolism of lipids, glucose, and anti-inflammatory cytokines. In fact, many studies suggest that PPARγ plays a key role in the ability of n-3 PUFA, specifically DHA, to reduce inflammation [25,26,27]. Additionally, an experimental study raised the hypothesis that n-3 fatty acids stimulate the secretion of adiponectin in epididymal fat in a PPARγ-dependent and PPARα-independent way [9]. Consequently, it is believed that part of the anti-inflammatory association observed for n-3 fatty acid in our study is mediated by these mechanisms.

Furthermore, DHA can reduce the production of eicosanoids derived from arachidonic acid (AA) by competing with it to be incorporated into the phospholipids of the cell membrane, reducing inflammation [28]. Finally, it can act directly on inflammatory cells through membrane receptors, such as GPR120 (G-protein coupled receptor 120), reducing the expression of nuclear factor kappa B (NF-kB) in macrophages and, consequently, the production of inflammatory cytokines, such as tumor necrosis factor alpha (TNF-α) and Toll-like receptors (TRLs) [2,29]. This molecular mechanism is complementary to the PPARs pathway to explain the association between n-3 and the Adipo/Lep ratio. 

The findings of the present study reinforce n-3 fatty acid as an inflammation modulator agent and highlight the importance of its consumption in reducing inflammation, including in severe obesity. Several studies show that the Westernized diet, characteristic of the current diet patterns, represents an increase in the consumption of calories in saturated fat and n-6 fatty acids, with a reduction in the intake of n-3 fatty acids. The general population has a low intake of n-3 fatty acids and is unable to reach the recommended levels of n6/n3 [30,31]. Thus, public policies that favor the consumption of foods that are sources of n-3 [29,32], especially in countries where the consumption of these foods is low, can help control inflammation and minimize unfavorable clinical outcomes resulting from obesity.

On the other hand, the regression analyses demonstrated C20:3 n-6 fatty acid as a predictor factor for the Lep/Adipo ratio, revealing the relationship of this type of fatty acid in the pro-inflammatory process. Linoleic fatty acid is a precursor for arachidonic acid (AA), leading to the formation of eicosanoids. These eicosanoids subsequently give rise to series 2 prostaglandins, series 4 leukotrienes, and related metabolites that collectively regulate pro-inflammatory activities, including the production of pro-inflammatory cytokines [32,33]. Considering that, in high concentrations, leptin is also a pro-inflammatory mediator, DGLA corroborates to a more pronounced state of inflammation in obesity [34].

In fact, our study observed a positive correlation between CRP and 18:3 n-6 (*p* = 0.01), a precursor of AA. Positive correlations between CRP and 20:4 n-3 and 18:3 n-3 were also observed in our study, which are two fatty acids that can be converted in n-6 fatty acids depending on the desaturase’s concentration [35].

It appears that the increased intake of precursors, such as AA, increases their content in the cell membrane, which may lead to the increased production of pro-inflammatory eicosanoids. Although dietary intake was not assessed in this study, it is known that a Westernized diet [36] can contribute to an increase in AA precursor consumption. This can be found in vegetable oils and derived products, such as margarine [37]. Older studies demonstrated that high levels of AA have a strong association with the formation of substances that aggregate platelets, such as thromboxanes [37,38]. In any case, investigating the consumption of foods that can increase arachidonic acid levels is important for the management of obesity, mainly from the perspective of inflammation [33].

In the present study, 21 fatty acids were identified in the volunteers’ plasma, including AGS, AGMI, and PUFA (Table 1), similarly to a previous study on the topic. Although we do not have a control group, it has been previously demonstrated by Bermúdez-Cardona and Velásquez-Rodríguez (2016) that only the DHA fatty acid differed between women with obesity and normal weight, and others reinforce that DHA might play an important role in obesity and cardiovascular disease [38,39]. Thus, n-3 supplementation can be an ally in controlling chronic low-grade inflammation in this population, either by promoting the anti-inflammatory or reducing the pro-inflammatory pathways, or both [2]. Albracht-Schulte et al. [2] suggest that omega-3 fatty acid can be an adjuvant in the treatment of obesity and metabolic syndrome, along with lifestyle modifications and pharmacotherapy. However, genetic and epigenetic factors require further studies, as they may interfere with the outcomes of greater n-3 consumption [33].

Even with the limitations of the study, such as the small sample size, the lack of a control group, and the lack of information about the volunteers’ food consumption, it was possible to observe the association between unsaturated fatty acids and the inflammatory profile. No saturated fatty acids were correlated with pro- and anti-inflammatory markers, which suggests that the strategy should be focused on unsaturated fatty acids. Furthermore, this is the first study carried out only with women with severe obesity, a condition that is increasing in our society. More studies are needed with a larger sample, evaluating dietary intake, to better understand the influence of food on circulating fatty acids.

## 5. Conclusions

The fatty acid DHA was shown to be a positive predictor for the anti-inflammatory adipo/lep ratio, whereas C20:3 n-6 was a predictor for inflammatory markers in women with severe obesity. The data suggest an intrinsic relationship between polyunsaturated fatty acids and inflammation in obesity, emphasizing the importance of public strategies to encourage the greater consumption of series n-3 fatty acids and to reduce the consumption of series n-6 fatty acids in individuals with severe obesity.

## Figures and Tables

**Table 1 biomedicines-12-02248-t001:** Descriptive analysis of anthropometric data, serum fatty acids, and adipokines from women with severe obesity (n = 41).

	Mean/Median	SD/Quartile Range
Characteristics and Anthropometric Data		
Age (y)	40.22	±8.3
Height (m)	1.59	±0.05
Body mass (kg)	118.80	111.95–129.10
BMI (kg/m^2^)	46.99	42.97–51.90
Waist circumference (cm)	131.46	±12.52
Hip circumference (cm)	142.50	136.1–152.25
Ratio Waist circumference/Hip	0.90	±0.07
Fatty Acids (% by area)		
Saturated (SFA)		
TOTAL	56.19	±5.63
C14:0	4.26	±1.45
C16:0	15.61	±4.75
C20:0	0.55	±0.14
C22:0	0.20	0–0.25
Monounsaturated (MUFA)		
TOTAL	17.15	±2.99
C14:1C	3.17	±1.01
C16:1n7	0.82	0.72–1.08
C18:1n9	11.23	±3.92
C18:1n7	1.11	±0.35
C20:1n9	0.75	±0.22
TOTAL	17.15	+2.99
Polyunsaturated (PUFA)		
TOTAL	26.64	+4.04
Omega-6		
TOTAL	21.68	+4.41
C18:2n6	14.47	±5.16
C18:3n6	3.45	±1.05
C20:2n6	1.73	±0.69
C20:3n6	0.32	±0.99
C20:4n6	0.23	0–0.30
C22:2n6	1.15	0.87–1.44
Omega-3		
TOTAL	4.93	3.35–6.80
C18:3n3	2.53	1.82–3.46
C18:4n3	0.46	0.36–0.57
C20:3n3	0.48	0.37–0.78
C20:4n3	0.29	0.23–0.35
C20:5n3	0.08	±0.32
C22:6n3	0.45	0.29–0.56
Ratio SFA/PUFA	2.02	1.80–2.41
Ratio SFA/MUFA	3.41	±0.89
Ratio n3/n6	0.18	0.13–0.33
Ratio n6/n3	5.39	3.02–7.38
Inflammatory Markers		
CRP	0.95	0.42–1.44
Adiponectin (µg/dL)	7.67	5.76–11.42
Leptin (ng/dL)	32.23	28–44.71
Ratio Adiponectin/Leptin	0.21	0.14–0.29
Ratio Leptin/Adiponectin	4.78	3.41–6.92

BMI: Body mass index; Waist circumference (<80 cm); Ratio circumference Waist/Hip (≤0.8); CRP: C-reactive protein (<1 mg/dL); Adiponectin (>7 µg/dL); Leptin (31.7 ng/dL).

**Table 2 biomedicines-12-02248-t002:** Multiple linear regression model for Adipo/Lep ratio in women with severe obesity.

		β	*p*-Value	OR (95% IC)
Step 1	Age	0.021	0.010	1.021 (0.02–3.27)
	BMI	−0.010	0.609	0.989 (0.95–1.03)
	WC	0.001	0.837	1 (0.988–1.01)
	HC	0.005	0.569	1 (0.98–1.02)
	NC	−0.005	0.781	0.994 (0.95–1.03)
	C22:6n3	0.755	0.000	2.129 (1.740–2.604)
Step 2	Age	0.022	0.006	1 (1.007–1.037)
	BMI	−0.009	0.633	0.991 (0.954–1.028)
	HC	0.005	0.545	1 (0.987–1.024)
	NC	−0.005	0.793	0.994 (0.956–1.034)
	C22:6n3	0.754	0.000	2.127 (1.744–2.594)
Step 3	Age	0.022	0.004	1 (1.007–1.037)
	BMI	−0.010	0.551	0.989 (0.955–1.024)
	HC	0.006	0.517	1 (0.988–1.024)
	C22:6n3	0.755	0.000	2.129 (1.750–2.589)
Step 4	Age	0.021	0.004	1 (1.007–1.036)
	HC	0.001	0.790	1 (0.992–1.010)
	C22:6n3	0.761	0.000	2.142 (1.766–2.598)
Step 5	Age	0.020	0.003	1.020 (1.007–1.034)
	C22:6n3	0.760	0.000	2.139 (1.768–2.588)

BMI: Body mass index; WC: Waist circumference; HC: Hip circumference; NC: Neck circumference.

**Table 3 biomedicines-12-02248-t003:** Multiple linear regression model for Lep/Adipo ratio in women with severe obesity.

		β	*p*-Value	OR (95% IC)
Step 1	Age	−0.166	0.328	0.847 (0.698–1.176)
	BMI	−0.116	0.786	0.890 (0.384–2.058)
	WC	0.043	0.777	1 (0.774–1.408)
	HC	−0.052	0.795	0.949 (0.641–1.405)
	NC	0.700	0.120	2 (0.850–4.774)
	C20:3n6	5.215	0.000	184.053 (113.090–298.966)
Step 2	Age	−0.149	0.335	0.861 (0.630–1.162)
	BMI	−0.201	0.469	0.817 (0.476–1.402)
	WC	0.716	0.104	1 (0.775–1.393)
	NC	0.038	0.796	2 (0.881–4.756)
	C20:3n6	5,232	0.000	187.243 (120.015–292.130)
Step 3	Age	−0.143	0.343	0.866 (0.646–1.160)
	BMI	−0.157	0.465	0.854 (0.563–1.297)
	NC	0.735	0.087	2 (0.918–4.736)
	C20:3n6	5.364	0.000	213.575 (170.462–267.592)
Step 4	Age	−0.122	0.343	0.885 (0.665–1.177)
	NC	0.592	0.087	1.809 (0.877–3.728)
	C20:3n6	5.021	0.000	151.640 (145.542–157.994)
Step 5	NC	0.648	0.081	1.911 (0.941–3.883)
	C20:3n6	4.848	0.000	127.495 (12.791–1270.826)

BMI: Body mass index; WC: Waist circumference; HC: Hip circumference; NC: Neck circumference.

## Data Availability

The original contributions presented in this study are included in this article. Further inquiries can be directed to the corresponding authors.

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
