# Peer review of "Serum Fatty Acids and Inflammatory Patterns in Severe Obesity: A Preliminary Investigation in Women"

_biomedicines, 2024, doi:10.3390/biomedicines12102248_

Round 1

Reviewer 1 Report

Comments and Suggestions for Authors

This manuscript examined the relationship between serum fatty acids and inflammatory parameters in women with severe obesity. Although the topic may be of interest, I have some concerns about the conceptualization and main results.

On the one hand, the use of two ratios which are the same but in an inverse way (leptin/adiponectin and adiponectin/leptin ratios) and not observing similar correlations with the same fatty acids in the opposite direction, apart of being redundant, it raises serious doubts about these findings.  “positive correlation between the leptin/adiponectin ratio and the 20:3n6 fatty acid (r = 0.52, p = 0.001), and a similar positive correlation between the adiponectin/leptin ratio and 22:6n3 fatty acid (r = 0.74, p = 0.001).”

On the other hand, using a regression analysis without including covariates (like age) doesn’t provide much more information than a correlation analysis. Thus, overall, the results are not of great relevance, taking into account that they consist of one descriptive table, two tables of correlations with those ratios and one regression graph.

Another point is the lack of other inflammatory parameters, like CRP, cytokines, etc. Additionally, the introduction should be improved (it seems more like the discussion section) with a better contextualization. Finally, in general, this manuscript is immature and lacks many important points. I encourage the authors to refocus the general approach and carry out deeper research to get a manuscript of real relevance.

Comments on the Quality of English Language

English language could be improved, but is not the major issue

Author Response

reviewer's comments

This manuscript examined the relationship between serum fatty acids and inflammatory parameters in women with severe obesity. Although the topic may be of interest, I have some concerns about the conceptualization and main results.

On the one hand, the use of two ratios which are the same but in an inverse way (leptin/adiponectin and adiponectin/leptin ratios) and not observing similar correlations with the same fatty acids in the opposite direction, apart of being redundant, it raises serious doubts about these findings.  “positive correlation between the leptin/adiponectin ratio and the 20:3n6 fatty acid (r = 0.52, p = 0.001), and a similar positive correlation between the adiponectin/leptin ratio and 22:6n3 fatty acid (r = 0.74, p = 0.001).”

Dear reviewer, thank you very much for your comments and the opportunity to improve our manuscript. Regarding the ratios Leptin/ adiponectin and Adiponectin/leptin, it has been widely used together in the population with obesity to explore systemic inflammation, being the first one pro-inflammatory and the second one anti-inflammatory, as you can see in the references below. Although the ratios come from the same variables, they generate different values as you can observe in table 1 ( adipo/lep = 0.21; lep/adipo = 4.78). In fact, when we did a correlation analysis between these 2 ratios it was observed a correlation of p=0,094 r= - 0,265, which demonstrates they are not the same thing, otherwise the r would be 1.

Seabra da Silva SMC, de Piano Ganen A, Masquio DCL, Dos Santos Quaresma MVL, Estadella D, Oyama LM, Tock L, de Mello MT, Dâmaso AR, do Nascimento CMDPO, Caranti DA. The relationship between serum fatty acids and depressive symptoms in obese adolescents. Br J Nutr. 2022

Masquio DC, de Piano A, Campos RM, Sanches PL, Carnier J, Corgosinho FC, Netto BD, Carvalho-Ferreira JP, Oyama LM, Oller do Nascimento CM, Tock L, de Mello MT, Tufik S, Dâmaso AR. Reduction in saturated fat intake improves cardiovascular risks in obese adolescents during interdisciplinary therapy. Int J Clin Pract. 2015

de Carvalho-Ferreira JP, Masquio DC, da Silveira Campos RM, Dal Molin Netto B, Corgosinho FC, Sanches PL, Tock L, Tufik S, de Mello MT, Finlayson G, Dâmaso AR. Is there a role for leptin in the reduction of depression symptoms during weight loss therapy in obese adolescent girls and boys? Peptides. 2015

Netto BD, Bettini SC, Clemente AP, Ferreira JP, Boritza K, Souza Sde F, Von der Heyde ME, Earthman CP, Dâmaso AR. Roux-en-Y gastric bypass decreases pro-inflammatory and thrombotic biomarkers in individuals with extreme obesity. Obes Surg. 2015 Jun;25(6):1010-8.

On the other hand, using a regression analysis without including covariates (like age) doesn’t provide much more information than a correlation analysis. Thus, overall, the results are not of great relevance, taking into account that they consist of one descriptive table, two tables of correlations with those ratios and one regression graph.

Dear reviewer, we completely agree with you and we have decided to do a multivariate regression with the covariates and present it instead of the correlations, as a more robust statistical analysis. Thank you for helping us to improve the manuscript.

Another point is the lack of other inflammatory parameters, like CRP, cytokines, etc.

Dear reviewer, we haven't included the CRP in the analysis considering it is a marker of acute inflammation rather than systemic inflammation. Since we have this data we decided to include it. In fact, the mean value of CRP of the patients are within the normality range. However, when we performed correlation analyses we found interesting results. Thank you for the suggestion.

Additionally, the introduction should be improved (it seems more like the discussion section) with a better contextualization.

We have improved the introduction section to highlight the lack in the literature that we are aiming to answer. Thank you.

Finally, in general, this manuscript is immature and lacks many important points. I encourage the authors to refocus the general approach and carry out deeper research to get a manuscript of real relevance.

Thank you for your consideration. We hope the manuscript has reached a version suitable for the publication. Please, feel free to make any other considerations to improve the manuscript.

Comments on the Quality of English Language

English language could be improved, but is not the major issue

Thank you, we reviewed the entire manuscript.

Reviewer 2 Report

Comments and Suggestions for Authors

I have reviewed the manuscript entitled “Serum Fatty Acids and Inflammatory Patterns in Severe Obesity: A Preliminary Investigation in Women”. The study is interesting, it reports the relationship between the adiponectin/leptin and leptin/adiponectin ratios and a particular fatty acid in obese women.

In my opinion, the manuscript could be published after revision. My comments are as follows:

Introduction

1.- Include information regarding the physiological meaning of adipo/lep and lep/adipo ratios.

2.- Are obese people deficient in omega3 fatty acids?

Materials and Methods

1.- Why weren’t lean women included as a control group?

Results and discussion

1.- Change 46,99 for 46.99, line 115.

2.- In Table 1, why were some parameters expressed as mean ± SD, i.e., waist circumference, and others as mean (or median, this is not clear) and quartile ranger, i.e., hip circumference? Same for fatty acids’ %?

3.- The % of fatty acids observed in serum is characteristic of obese people? It would be interesting to compare it with the % in lean persons.

4.- The levels of adipokines found in serum are normal, high, low?

5.- Change 31,7 for 31.7, line 124.

6.- Why is lep/adipo ratio positively correlated with 20:3 n6? What would it be expected for lean people? Why no correlation was observed with 20:4 n6, which is more physiologically important?

7.- Why is adipo/lep ratio positively correlated with 22:6 n3? What would it be expected for lean people? The % of 22:6 n3 is low, high?

8.- Give more information about DHA being a biomarker of the inflammatory state of obesity.

9.- Change saturated for unsaturated, line 189.

10.- Change “cells” for “molecules”, line 204.

11.- Remove sentence lines 231-232.

Author Response

reviewer's comments

Introduction

1.- Include information regarding the physiological meaning of adipo/lep and lep/adipo ratios.

Dear reviewer, we have included that in the introduction section.

2.- Are obese people deficient in omega3 fatty acids?

Dear reviewer, the literature shows that the Westernized diet, characteristic of the current diet, represents an increase in the consumption of calories in saturated fat and omega-6 fatty acids, with a reduction in the intake of omega-3 fatty acids. The general population has a low intake of omega-3 fatty acids and is unable to reach the recommended levels of n6/n3. Since the serum fatty acids is an indirect marker of food consumption we can not assume 100% that they were insufficient, but it is expected to be low due to the diet pattern of this population. We have included in the discussion section a paragraph about it.

  • Candela, CG.; López, LMB.; Kohen, LV. Importance of a balanced omega 6/omega 3 ratio for the maintenance of health. Nutritional recommendations. Nutr Hosp, 2011; 26(2): 323-329.
  • Simopoulos AP. The importance of the ratio of omega-6/omega-3 essential fatty acids. Biomed Pharmacother. 2002 Oct;56(8):365-79. doi: 10.1016/s0753-3322(02)00253-6. PMID: 12442909.
  • Simopoulos AP. Evolutionary aspects of diet, the omega-6/omega-3 ratio and genetic variation: nutritional implications for chronic diseases. Biomed Pharmacother. 2006 Nov;60(9):502-7. doi: 10.1016/j.biopha.2006.07.080. Epub 2006 Aug 28. PMID: 17045449..

Materials and Methods

1.- Why weren’t lean women included as a control group?

Dear reviewer, thank you for this important observation. We recognize that our study focused on a specific group of individuals, specifically women with severe obesity. This decision was based on a few factors. We aim to address the increasing prevalence of severe obesity among women and its potential inflammatory markers related to fatty acids. In addition, we understand that the exclusion of a control group limits the generalizability of our results. However, the main objective of this study was to understand whether the relationship between markers of inflammation and fatty acids could be crucial for systemic inflammation. However, we have included this as a limiting factor.

Although this data is limited to women with severe obesity before bariatric surgery,  this data highlights the importance of fatty acids in inflammation in this group, which can give light to public health measurements to facilitate the access in omega-3 rich foods for the population.

Results and discussion

1.- Change 46,99 for 46.99, line 115. Ok

2.- In Table 1, why were some parameters expressed as mean ± SD, i.e., waist circumference, and others as mean (or median, this is not clear) and quartile ranger, i.e., hip circumference? Same for fatty acids’ %?

Data are presented as means and standard deviation or median and quartile range according to data normality tested by the Shapiro Wilk test.

3.- The % of fatty acids observed in serum is characteristic of obese people? It would be interesting to compare it with the % in lean persons.

In the present study, 21 fatty acids were identified in the volunteers' plasma, including AGS, AGMI and PUFA (Table 1), similarly to a previous study on the topic. Furthermore, one study compared the fatty acids profile between people with obesity and normal weight and they only found differences in the Dihomo-gamma-linolenic fatty acid.

Bermúdez-Cardona J, Velásquez-Rodríguez C. Profile of Free Fatty Acids and Fractions of Phospholipids, Cholesterol Esters and Triglycerides in Serum of Obese Youth with and without Metabolic Syndrome. Nutrients. 2016 Feb 15;8(2):54. doi: 10.3390/nu8020054. PMID: 26891317; PMCID: PMC4772025.

Kim OY, Lim HH, Lee MJ, Kim JY, Lee JH. Association of fatty acid composition in serum phospholipids with metabolic syndrome and arterial stiffness. Nutr Metab Cardiovasc Dis. 2013 Apr;23(4):366-74. doi: 10.1016/j.numecd.2011.06.006. Epub 2011 Sep 15. PMID: 21920716.

4.- The levels of adipokines found in serum are normal, high, low?

The reference values ​​found in the literature are for women with a healthy weight. In order to seek an approximate reference for the population of our study, we used references from a group that studies a population similar to ours.

  • Farias, G.; Molin Netto, B dal.; Boritza, K.; Bettini, SC.; Vilela, RM.; Dâmaso, AR. Impact of Weight Loss on Inflammation State and Endothelial Markers Among Individuals with Extreme Obesity After Gastric Bypass Surgery: a 2-Year Follow-up Study. Obes Surg, 2020; 30(5): 1881-1890.

5.- Change 31,7 for 31.7, line 124. Ok

6.- Why is lep/adipo ratio positively correlated with 20:3 n6? What would it be expected for lean people? Why no correlation was observed with 20:4 n6, which is more physiologically important?

It is known that plasma fatty acids do not represent only dietary intake, as they are affected by genetic factors.

  • Hodson L, Skeaff CM, Fielding BA. Fatty acid composition of adipose tissue and blood in humans and its use as a biomarker of dietary intake. Prog Lipid Res 2008;47:348–80. https://doi.org/10.1016/J.PLIPRES.2008.03.003.

7.- Why is adipo/lep ratio positively correlated with 22:6 n3? What would it be expected for lean people? The % of 22:6 n3 is low, high?

The adiponectin/leptin ratio is anti-inflammatory for people with obesity and without obesity, but lean people are not expected to have the low-grade inflammation that occurs in obesity.

8.- Give more information about DHA being a biomarker of the inflammatory state of obesity.

- Ligiang et.al, 2023 identified DHA as a potential biomarker of obesity and Siriwardhana et. al, 2013 states that DHA confers anti-inflammatory effects through several mechanisms, such as the activation of AMPK and PPARy.

  • Liqiang S, Fang-Hui L, Minghui Q, Yanan Y, Haichun C. Free fatty acids and peripheral blood mononuclear cells (PBMC) are correlated with chronic inflammation in obesity. Lipids Health Dis. 2023; 22 : 1–9.
  • Siriwardhana, N.; Kalupahana, NS.; Cekanova, M.; LeMieux M.; Greer, B.; Moustaid-Moussa, N. Modulation of adipose tissue inflammation by bioactive food compounds. JNB, 2013; 24(4):613-623.

9.- Change saturated for unsaturated, line 189. Ok

10.- Change “cells” for “molecules”, line 204. Ok

11.- Remove sentence lines 231-232. Ok

Round 2

Reviewer 1 Report

Comments and Suggestions for Authors

Authors have carried out changes in the manuscript considering my previous comments. Some improvements, like providing some figures instead of tables or a deeper discussion, could be made. But, overall, the current manuscript has improved considerably.

Comments on the Quality of English Language

Quality of English is acceptable

Author Response

Dear reviewer, thank you very much for your comments. We made some pertinent changes to the discussion. Our statistician suggested leaving a table and not a regression graph (figure), as this type of graph with three variables would not be interesting. Thanks again.